**Data Availability Statement:** All relevant data are available within the paper.

**Funding:** This national survey was financially supported by Ethiopian Food and Drug Authority

# A three years antimicrobials consumption in Ethiopia from 2017 to 2019: A cross- sectional study

**Million Tirfe[1], Asnakech Alemu[1], Wondie Alemu[1], Mengistab Woldearegay[2], Getachew Asfaw[3], Heran Gerba[4], Duru Kadi[1], Atalay Mulu Fentie[ID][5]***

1 Product Safety Directorate, Ethiopian Food and Drug Authority, Addis Ababa, Ethiopia, 2 World Health Organization Country Office for Ethiopia, Addis Ababa, Ethiopia, 3 Pharmaceutical and Medical Equipment Directorate, Ministry of Health, Addis Ababa, Ethiopia, 4 Director General, Ethiopian Food and Drug Authority, Addis Ababa, Ethiopia, 5 School of Pharmacy, College of Health Sciences, Addis Ababa University, Addis Ababa, Ethiopia

* atalay.mulu@aau.edu.et

## Abstract

### Background

The widespread use and misuse of antimicrobials are the major driving factor for antimicrobial resistance (AMR) that threatens the health of human beings globally. Thus, monitoring antimicrobial consumption at national level is crucial to prevent and contain AMR. Nevertheless, there is no well-established system for recording and reporting of antimicrobial consumption in Ethiopia. Hence, the national antimicrobial consumption survey was conducted to generate evidence for decision-making on the appropriate use of antimicrobials in Ethiopia and tackle AMR.

### Methods

All imported and locally manufactured antimicrobials from 2017 to 2019 were from the Ethiopian Food and Drug Authority and local manufacturers database, respectively. Data were collected and analyzed descriptively in accordance with the World Health Organization (WHO) Anatomical Therapeutic Chemical (ATC) and defined daily doses (DDD) classification and methodology.

### Results

The average DDD/1,000 inhabitants for all antimicrobials was 15.36. The DDD/1,000 inhabitants fell down sharply from 37.03 in 2017 to 4.30 in 2018, before slightly rising to 4.75 in 2019. The majority of the consumed antimicrobials were comprised of oral antimicrobials (98.6%), while parenteral antimicrobials made up 1.4%. Tetracyclines (35.81%), fluoroquinolones (20.19%), macrolides (13.92%), antiretrovirals (10.57%), and cephalosporins (9.63%) were the most frequently consumed classes of antimicrobials during the three years period. About 75.83% of the consumed antimicrobials fall under the WHO AWaRe classification and 67.87% of antimicrobial consumption was from the WHO Access class

through the annual government budget allocated specifically for the consumption survey.

**Competing interests:** The authors have declared that no competing interests exist

medications, with Watch and Reserve classes accounting for 32.13% and <1%, respectively. Similarly, about 86.90% of the antimicrobials fall under the Ethiopian AWaRe classification, with Access, Watch, and Reserve accounting for 87.73%, 12.26%, and <1%, respectively.

## Conclusion

Due to the peculiarities of our settings, our findings may have some similarities and differences with similar studies from other countries. Hence, we recommend for all concerned bodies to work collaboratively to improve monitoring of antimicrobial consumption at different levels of the Ethiopian healthcare tier system. Future work is necessary to establish a strong system of reporting of antimicrobial consumption patterns in Ethiopia.

## Introduction

Antimicrobials have saved millions of lives, substantially reduced the burden of infectious diseases, improved quality of life, helped increase life expectancy starting from penicillin's first discovery by Alexander Fleming in 1928 [1]. Since then, alarmingly increasing rational and irrational use of antimicrobials has led to consumption-driven antimicrobial resistance (AMR). Antimicrobial resistance is a major threat to health and human development, affecting our ability to treat a range of infections as well as other disease conditions [2].

Global antimicrobial consumption (AMC) has increased by 65% from 2000 to 2015. The highest AMC rates were reported in high-income countries (HICs) (France, New Zealand, Spain, Hong Kong, and the United States) in 2000. On the contrary, in 2015, four of the six countries with the highest consumption rates were from the Low-and Middle-Income Countries (LMICs) (Turkey, Tunisia, Algeria, and Romania). Although the total AMC has increased by 6% between 2000 and 2015 in HICs, there has been a 4% decline in the AMC rate. However, in LMICs, both the total and rate of AMC have increased significantly by 114.4% and 77%, respectively [3, 4].

The worldwide AMC rate of broad-spectrum penicillin's, the most commonly consumed class of antibiotics, increased by 36% between 2000 and 2015. The highest increment was reported in LMICs (56%), compared to HICs (15%). Whereas, the consumption rate of the next three most commonly consumed classes of antimicrobials, cephalosporin (20%), quinolones (12%), and macrolides (12%), has generally decreased in HICs but significantly increased in LMICs. Cephalosporins, quinolones, and macrolides consumption rates in LMICs have increased by 399%, 125%, and 119%, respectively, while consumption rates in HICs have decreased by 18%, 1%, and 25%, respectively [4].

Although the consumption and use of antimicrobials is increasing from time to time, diseases due to drug-resistant infections alone kill more than 1.27 million people per year [5]. If no action is taken, by 2050, 10 million people per year will die because of AMR alone. Furthermore, AMR will cost up to 100 trillion USD by 2050, and 28 million people could fall into extreme poverty, while world GDP will fall by 2–3.5% [6].

Antimicrobial resistance is a complex problem with interrelated and vast causes. However, the core contributors to the spread of AMR are systematic misuse and exuberant use of antimicrobials in humans, animals, and food production sectors [7, 8]. There is evidence showing a positive linear relationship between irrational use of antimicrobials and the emergence of

resistant microbes [9–12]. Therefore, optimizing antimicrobial use is the best solution to minimize and/or contain AMR. Hence, this study was intended to measure AMC in Ethiopia between 2017 and 2019.

# Methods

## Study design

This was a retrospective cross-sectional investigation of Ethiopian Food and Drug Authority (EFDA) legally registered AMC for human health sector as per imported and locally manufactured antimicrobials all over Ethiopia from 2017 to 2019 (a three years period).

## Study area

This AMC survey was conducted in Ethiopia, a country located in the horn of Africa. Ethiopia lies completely within the tropical latitudes and is relatively compact, with similar North-South and East-West dimensions. Ethiopia is the largest and the second-most populated country in Africa (Table 1) [13, 14]. The Ethiopian Health System, guided by the National Health Policy, has a three-tier system of primary, secondary, and tertiary level healthcare [14]. Health care services are mostly provided by the public sector, with some services provided by the private sector, non-governmental organizations, and faith-based organizations. Infectious diseases and nutritional disorders are the major health problems of the Ethiopian population. This implies antimicrobials take the major share in health expenditure [15].

## Source of data

Data on all antimicrobials imported into Ethiopia was obtained from the Ethiopian Food and Drug Authority (EFDA) import database. The EFDA regulates and approves all medicines and supplies, including antimicrobials that are imported into the country by public and private importers. The products are entered based on various categories (e.g., medicines, medical supplies) with details such as product strength, form, pack size, and total quantity. Moreover, additional AMC data was obtained from all the Ethiopia-based manufacturers that produce antimicrobials for local consumption (during the study there were a total of 12 different local manufactures registered to manufacture antimicrobials in Ethiopia). All data sources used in this study were publicly available and anonymized.

## Eligibility criteria

All antimicrobials for systemic use (oral and parenteral preparations) imported and locally manufactured in Ethiopia were included in the study. Anthelmintics and antimicrobials used for local therapy (e.g., topical ointments, ear/eye drops) were excluded.

## Data extraction and categorization

The information was gathered and analysed in accordance with the WHO Anatomical Therapeutic Chemical (ATC) and Defined Daily Dose(DDD) Methodology [15]. The AMC tool was

**Table 1. Ethiopian population projection [13, 14].**

| Year | Population |
|------|------------|
| 2017 | 94,228,814 |
| 2018 | 98,556,080 |
| 2019 | 100,684,575 |

used to assign an ATC Code and calculate DDD [16]. The study included all eligible legally imported and locally manufactured antimicrobials from a three years period categorized under the subgroup A07 (alimentary anti-infectives), D01 (Antifungals for systemic uses), J01 (Antibacterial for systemic use), J02 (Antimycotics for systemic use), J04 (Antimycobacterial for treatment of tuberculosis and leprosy), J05 (All Antivirals) and P01 (Nitroimidazoles, Anti-protozoals) of the ATC classification system, including parenteral and oral formulations.

### Data collection and entry

Data on imported antimicrobials was collected from various ports of entry for each year using a standardized excel based data abstraction format. For the year 2017 and 2018, data was abstracted from a paper-based import databases of EFDA containing any eligible antimicrobial agent whereas for the year 2019 data was extracted from electronic regulatory information system of EFDA (the electronic regulatory information system was fully functional since 2019). For the local manufacturers' antimicrobial data, the EFDA officially requested the 10 local manufacturers to send their antimicrobial manufacturing data (already manufactured and sold to distributors) as per the already developed standardized data abstraction format for the stated years. Data from the hard copy was entered into the Excel based standardized data abstraction format adopted from WHO AMC data collection tool. As a result, the following data variables were collected: population by year; name of substance; strength of active ingredient(s); route of administration; dosage form; and quantity. Data were manually cleared by reviewing each item and validated using the WHO tool (Macros) (Fig 1).

### Data analysis

All collected data on imported and locally manufactured antimicrobials were entered into the WHO AMC tool. The AMC tool also automatically generated the consumption into DDD format and DDD per 1000 inhabitants. The latter was calculated manually using population estimates for 2017–2019. For the analysis, totals and summary statistics are presented for DDD per 1,000 inhabitants per day (DID) using the WHO tool. The products were thematized and consumption for the entire study period was therefore aggregated using the macro in the WHO data collection and analysis tool.

### Ethical considerations

A permission letter was obtained from EFDA general director and submitted to the EFDA ports of entry branch offices, the Medicines Registration and Licensing Directorate, and local manufacturers to obtain all the necessary data. Nine trained data collectors were recruited. Trained data collectors were used to collect data from the sites and were also advised to focus on aggregates, not on any business information about firms or products other than antimicrobials. Besides, there was continuous support from data collection supervisors at national level.

## Results

### Overall national antimicrobial consumption

The average total consumption of all antimicrobials in Ethiopia over three years (2017–2019) was 601,284,712 DDDs. The average DDD per DID for all antimicrobials was 15.3601 (37.0308, 4.3008, and 4.7488 for the year 2017, 2018 and 2019, respectively) showing dramatic reduction. The total DDD, DID for each year of the three years is summarized in the Table 2.

Tetracycline's (35.81%), fluoroquinolones (20.19%), macrolides (13.92%), antiretrovirals (10.57%), and cephalosporins (9.63%) were the most commonly consumed classes of

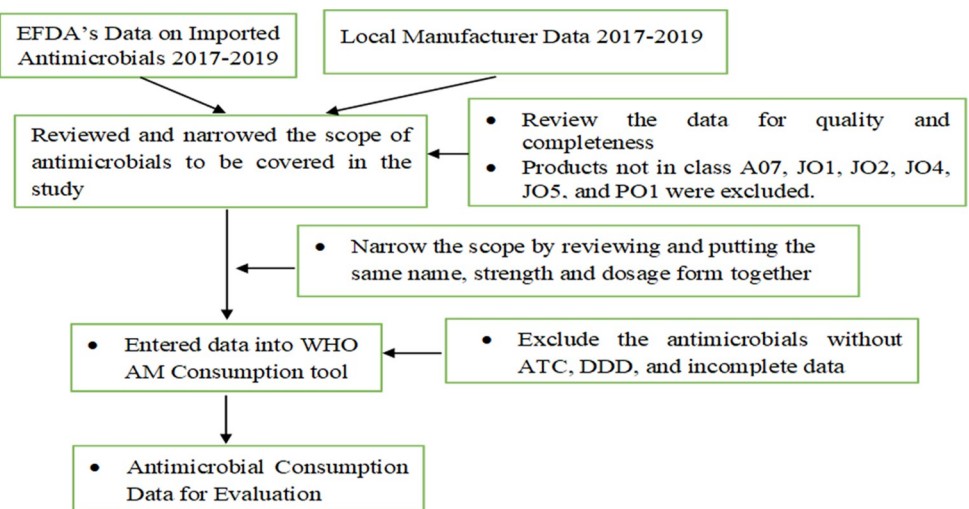

**Fig 1. Data collection and cleaning process for antimicrobial consumption analysis.**

antimicrobials accounting for 90.12% of the average DDDs and DID for three years (2017–2019) (Fig 2).

## Antimicrobial consumption in DID by ATC and by route of administration

As shown in Table 3, antibacterial for systemic use (J01) comprised the vast majority of AMC each year, accounting for 91.86%, 58.9%, and 74.77% of the consumption in 2017, 2018, and 2019, respectively. Hence, antibacterials for systemic use accounted for an average of 87.03%

**Table 2. Summary of antimicrobial consumption in defined daily dose by therapeutic category from 2017–2019.**

| Therapeutic category | 2017 | | | 2018 | | | 2019 | | | Average three | | | |
|---|---|---|---|---|---|---|---|---|---|---|---|---|---|
| | DDD | % | DID | DDD | % | DID | DDD | % | DID | DDD | % DDD | DID | % DID |
| Tetracyclines | 624682040 | 43.4372 | 16.0851 | 2259000 | 1.3175 | 0.0567 | 19084060 | 9.8236 | 0.4665 | 215341700 | 35.8136 | 5.5361 | 36.0420 |
| Fluoroquinolone | 341395592 | 23.7389 | 8.7907 | 4528766 | 2.6413 | 0.1136 | 18284100 | 9.4118 | 0.4469 | 121402819.3 | 20.1906 | 3.1171 | 20.2933 |
| Macrolides | 126203120.9 | 8.7755 | 3.2496 | 68746801.5 | 40.0954 | 1.7244 | 56184031.83 | 28.9209 | 1.3734 | 83711318.09 | 13.9221 | 2.1158 | 13.7747 |
| Antiretroviral | 104419740 | 7.2608 | 2.6887 | 42524618 | 24.8018 | 1.0667 | 43638508 | 22.4630 | 1.0667 | 63527622 | 10.5653 | 1.6074 | 10.4646 |
| Cephalosporins | 162422326.2 | 11.2940 | 4.1823 | 3942186 | 2.2992 | 0.0989 | 7342953.5 | 3.7798 | 0.1795 | 57902488.58 | 9.6298 | 1.4869 | 9.6801 |
| Penicillin | 59195303.3 | 4.1161 | 1.5242 | 20323277 | 11.8532 | 0.5098 | 40589750.23 | 20.8937 | 0.9922 | 40036110.18 | 6.6584 | 1.0087 | 6.5673 |
| Medicine for amoebiasis | 1340260 | 0.0932 | 0.0345 | 24972222.5 | 14.5646 | 0.6264 | 2927001.25 | 1.5067 | 0.0715 | 9746494.583 | 1.6209 | 0.2442 | 1.5895 |
| Anti-TB | 10877164.4 | 0.7563 | 0.2801 | 1128422.44 | 0.6581 | 0.0283 | 403247 | 0.2076 | 0.0099 | 4136277.947 | 0.6879 | 0.1061 | 0.6906 |
| Miscellaneous | 4672536.333 | 0.3249 | 0.1203 | 985917.667 | 0.5750 | 0.0247 | 3384846.083 | 1.7424 | 0.0827 | 3014433.361 | 0.5013 | 0.0759 | 0.4943 |
| Aminoglycosides | 2504142 | 0.1741 | 0.0645 | 223835.5 | 0.1305 | 0.0056 | 105471.4167 | 0.0543 | 0.0026 | 944482.9722 | 0.1571 | 0.0242 | 0.1577 |
| Antimalarial | 36514.28571 | 0.0025 | 0.0009 | 1443490.29 | 0.8419 | 0.0362 | 1081286.286 | 0.5566 | 0.0264 | 853763.619 | 0.1420 | 0.0212 | 0.1380 |
| Antifungal | 82857.14286 | 0.0058 | 0.0021 | 123145.467 | 0.0718 | 0.0031 | 1106421.819 | 0.5695 | 0.0270 | 437474.8095 | 0.0728 | 0.0108 | 0.0700 |
| Antileprotic | 296000 | 0.0206 | 0.0076 | 205600 | 0.1199 | 0.0052 | 132700 | 0.0683 | 0.0032 | 211433.3333 | 0.0352 | 0.0053 | 0.0348 |
| Other antivirals | 0 | 0.0000 | 0.0000 | 48849.75 | 0.0285 | 0.0012 | 2695 | 0.0014 | 0.0001 | 17181.58333 | 0.0029 | 0.0004 | 0.0028 |
| Carbapenem | 333.33333 | 0.0000 | 0.0000 | 2000 | 0.0012 | 0.0001 | 1000 | 0.0005 | 0.0000 | 1111.111111 | 0.0002 | 0.0000 | 0.0002 |
| Total | **1438127930** | 100 | 37.0308 | **171458132.1** | 100 | 4.3008 | 194268072 | 100 | 4.7488 | 601284712 | 100 | 15.3601 | 100 |

DDD: Defined daily dose, DID: Defined daily dose per 1000 inhabitants per day.

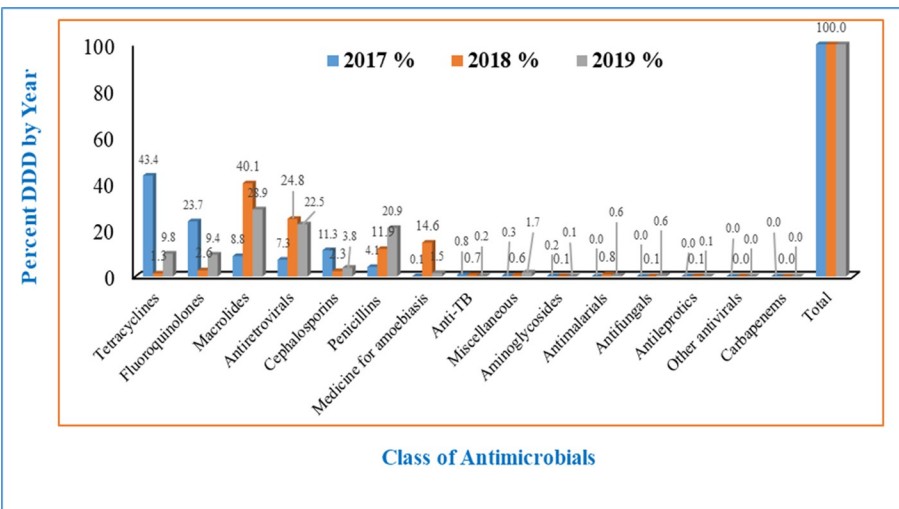

**Fig 2. Antimicrobial consumption by year and anatomical therapeutic chemical level.**

of the three years. Similarly, antiviral (JO5) comprises almost all antiretroviral consumption, each accounting for 7.26%, 24.8%, and 22.51% of consumption in 2017, 2018, and 2019, respectively. Overall, antivirals accounted for 10.47% of the three-year average consumption. Antiprotozoal (P01) consumption each year consists of imidazole and antimalarial consumption, accounting for 0.1%, 15.4%, and 2.07% of the consumption in 2017, 2018, and 2019, respectively.

## Consumption by AWaRe classification

Antimicrobials classified using WHO AWaRe (access, watch, and reserve) classification were the most commonly used drugs accounting for 76.2%, 72.5% and 75.8% of the consumption in the years 2017, 2018, and 2019, respectively. Similarly, the Ethiopian AWaRe Classification category of antimicrobials also comprised most of the AMC each year, accounting for 89.96%,

**Table 3. Total antimicrobial consumption per defined daily dose (DDD) per 1000 inhabitants per day according to the anatomical, therapeutic, and chemical (ATC) classification system and by route of administration.**

| Product | DDD/1000/day (DID) (%) | | | | | | Overall, three years consumption | | |
| --- | --- | --- | --- | --- | --- | --- | --- | --- | --- |
| | 2017 | | 2018 | | 2019 | | Mean | | SD |
| | Average | % | Average | % | Average | % | Average | % | |
| Total | 37.03 | 100 | 4.3 | 100 | 4.74 | 100 | 15.3567 | 100 | 15.3264 |
| J01 (Antibacterial for systemic use) all | 34.016 | 91.86 | 2.5326 | 58.9 | 3.544 | 74.77 | 13.3642 | 87.0252 | 14.6089 |
| • J 01 Oral | 33.527 | 90.54 | 2.444 | 56.84 | 3.474 | 73.29 | 13.1483 | 98.3845 | 14.4160 |
| • J01 Parenteral | 0.488 | 1.32 | 0.089 | 2.07 | 0.0699 | 1.47 | 0.2156 | 1.6398 | 0.1928 |
| J04 (antimycobacterial) all | 0.2885 | 0.78 | 0.0345 | 0.8 | 0.013 | 0.27 | 0.1120 | 0.8381 | 0.1251 |
| • J04 oral | 0.288 | 0.78 | 0.0334 | 0.78 | 0.013 | 0.27 | 0.1114 | 99.4643 | 0.1250 |
| • J04 Parenteral | 0.001 | 0 | 0.0012 | 0.03 | 0 | 0 | 0.0007 | 0.6250 | 0.0005 |
| J05 all (Antiviral) | 2.6887 | 7.26 | 1.0679 | 24.8 | 1.0668 | 22.51 | 1.6078 | 10.4697 | 0.7643 |
| • J05 oral | 2.6887 | 7.26 | 1.0679 | 24.83 | 1.0668 | 22.51 | 1.6078 | 100 | 0.7643 |
| P01 Antiprotozoal | 0.0355 | 0.1 | 0.6626 | 15.4 | 0.098 | 2.07 | 0.2653 | 1.7276 | 0.2821 |
| • P01AB (Nitroimidazole) | 0.0345 | 0.09 | 0.6264 | 14.6 | 0.0716 | 1.51 | 0.2442 | 92.0467 | 0.2707 |
| • P01B (All Antimalarial) | 0.0009 | 0 | 0.0362 | 0.84 | 0.0264 | 0.56 | 0.0212 | 7.9910 | 0.0149 |

72.5% and 75.8% of the consumption in 2017, 2018 and 2019, respectively. Antimicrobials of the access category are the most commonly consumed groups accounting for 67.87% and 87.73% of the consumption in both the WHO AWaRe Classification [17] and the Ethiopia AWaRe Classification [18] for the three years aggregate consumption, respectively (Table 4).

## Most commonly consumed antimicrobials by chemical substance group

As presented in Table 5, antibacterial in the J01 category of substances to ATC5 level are the most consumed class of antimicrobials from 2017 to 2019. Doxycycline, norfloxacin, azithromycin, ciprofloxacin, cefalexin, amoxicillin, cefixime, amoxicillin clavulanic acid, erythromycin and clarithromycin were the 10 most commonly consumed antibiotics, accounting 85.81% (DID 31.7741), 55.74% (DID 2.3973) and 71.54% (DID 3.3972) of the consumption in 2017, 2018, and 2019, respectively.

## Discussion

To our knowledge, this is the first comprehensive study on AMC conducted in Ethiopia and it is also one of a few studies on AMC in Sub-Saharan Africa region. It is a national study on AMC based on data on imported and locally manufactured antimicrobials and the findings of this study were already disseminated to Ministry of Health, healthcare facilities, medicine retail outlets and other relevant stakeholders in Ethiopia and internationally. The results of this study will serve as a baseline for future monitoring of AMC in Ethiopia.

The finding of the study showed that the overall mean consumption of antimicrobials over three years (2017–2019) in Ethiopia was 15.36 DID. This was lower than the results of a study from Tanzania where the average overall consumption over three years (2017–2019) was 80.80 DID [19] and European Union (EU) study's finding with 18.4 DID in 2018 [20]. However, the mean overall consumption of antimicrobials in Ethiopia over three years was higher than the Japan's study from 2004–2016 with 14.41 DID [21] and that of three years study conducted in China with 11.20 DID in 2015, 10.13 DID in 2016 and 12.99 DID in 2018 [22]. Higher use of antimicrobials in Ethiopia compared to Japan and China might be due to a higher burden of communicable illnesses, limited diagnostic and history/physical examination alone based diagnosis of infectious diseases fueling the empiric and irrational prescribing practice, widespread dispensing of antimicrobials without prescription and irrational use of antimicrobials for human health sectors [23–25].

The find of this study falls within in the range indicated by the WHO comprehensive study which states: though there is intra and inter-regional variation, the overall consumption of antibiotics ranged from 4.4 to 64.4 DID [26]. Majority of the antimicrobials consumed over three years (2017–2019) in Ethiopia were from the access category of both WHO and Ethiopia AWaRe Classification. Comparison of the percentage of access antibiotic consumption of Ethiopia and WHO AWaRe categorization, the percentage of access antibiotic consumption as per the Ethiopia AWaRe categorization was higher than that of WHO AWaRe categorization. This might be due to the fact that some antimicrobials categorized as watch in WHO AWaRe classification are categorized as access in Ethiopia AWaRe Categorization. Health facilities setups, diseases conditions, diagnostic facilities, professional capacities and making antimicrobials accessible to rural community in Ethiopia are considered in deciding the AWaRe class of antimicrobials and might affect the proportion. The vast majority of antibiotic consumption in Ethiopia (76.2%) was already as to and in accordance with the WHO's target to make at least 60% of antimicrobials consumed to be in Access component of the AWaRe Classification [17].

Doxycycline, norfloxacin, azithromycin, ciprofloxacin and cefalexin were the most consumed antibiotics in Ethiopia from 2017 to 2019 with mean DID of 5.54± 7.46, 1.90 ±2.69,

**Table 4. Antimicrobial consumption by AWaRe classification from 2017–2019 in Ethiopia.**

| Product category | DDD/1000/day (%) | | | | | | Overall, three years aggregate consumption | | |
|---|---|---|---|---|---|---|---|---|---|
| | 2017 | | 2018 | | 2019 | | Mean | | SD |
| | Average | % | Average | % | Average | % | Average | % | |
| **WHO AWaRe Classification** | 28.22 | 76.22 | 3.12 | 72.50 | 3.59 | 75.80 | 11.65 | 75.83 | 11.72 |
| Access | 20.68 | 55.84 | 1.28 | 29.80 | 1.75 | 36.98 | 7.90 | 67.87 | 9.03 |
| Watch | 7.55 | 20.38 | 1.84 | 42.80 | 1.84 | 38.82 | 3.74 | 32.13 | 2.69 |
| Reserve | 0.00 | 0.00 | 0.00 | 0.03 | 0.00 | 0.00 | 0.00 | 0.00 | 0.00 |
| **Ethiopia AWaRe Classification** | 33.31 | 89.96 | 3.12 | 72.50 | 3.60 | 75.98 | 13.34 | 86.90 | 14.12 |
| Access | 28.99 | 78.29 | 3.00 | 69.80 | 3.13 | 66.04 | 11.71 | 87.73 | 12.22 |
| Watch | 4.32 | 11.68 | 0.11 | 2.66 | 0.47 | 9.89 | 1.64 | 12.26 | 1.91 |
| Reserve | 0.00 | 0.00 | 0.00 | 0.02 | 0.00 | 0.04 | 0.00 | 0.01 | 0.00 |

1.74± 0.33, 1.20 ±1.34 and 1.02 ± 1.30, respectively. Despite significantly higher than the Ethiopian consumption, doxycycline was also the most frequently consumed antibiotic in Tanzania with mean DID ± SD of 20.01 ± 24.53 [19]. In Ethiopia, doxycycline consumption was very high in 2017 compared to 2018 and 2019 consumption. This over consumption in 2017 might be due outbreak of acute watery diarrhea in Ethiopia in 2017 that led to over consumption of antimicrobials such as doxycycline as clearly showing the over consumption of doxycycline on Table 4 and hence the government imported large quantities of doxycycline. Conversely, the most consumed antibiotic class in the Iraqi [27] and Sierra Leone [28] studies were extended spectrum penicillin's and metronidazole, respectively. These two drugs were not among the top five most commonly consumed antibiotics in Ethiopia. This might be due to limited access to extended spectrum penicillin's, differences in diseases causing microbes as well as status of resistance in these countries [29, 30]. Similar to other countries, the prevalence of AMR to these highly consumed antimicrobials is alarmingly increasing in Ethiopia particularly for gram-negative bacteria's with a >50% pooled prevalence of resistance and may jeopardize the clinical decision and clinical outcomes [31–33]. Hence, this study highlighted the consumption has to be further studied against the burden of the respective antimicrobials resistance pattern and needs a concerted effort to preserve the precious resources to save the life of humankind.

The findings of this study showed that in Ethiopia over the period of three years (2017–2019), almost all consumed antimicrobials were administrated orally accounting for 98.6% of the consumption. This seems ideal and the overall consumption of injectable antimicrobial was 1.4% that was much lower when compared with community's perception on injection and AMC studies undertaken in other countries such as Tanzania where the overall consumption of injectable antimicrobial was 2.4% [19]. The consumption trend in Ethiopia showed an overall decreasing pattern from the 2017 to 2019. The main factors for this decline might be improved diagnostic laboratories, a bit improved prescribing patterns, and improved awareness of risk of antimicrobial resistances, scarcity of hard currency to import antimicrobials and poor documentation of imported antimicrobials.

This study had several strengths including the use of import data and data from local manufacturers to estimate national-level consumption as recommended by WHO. The other important strength was the use of standardized methodology that permits comparisons over time and across countries. However, the study was not without limitations. Some of the limitations of this study include:

**Table 5. Most commonly consumed antimicrobials by chemical substance group in Ethiopia between 2017–2019.**

| Rank | Generic Name | ATC Code | DID | | | Mean ± SD |
|------|-------------|----------|------|------|------|-----------|
| | | | 2017 | 2018 | 2019 | |
| J01: Antibacterial for systemic use | | | | | | |
| 1. | Doxycycline | J01AA02 | 16.0851 | 0.0567 | 0.4665 | 5.5361± 7.4612 |
| 2. | Norfloxacin | J01MA06 | 5.7014 | 0.0002 | 0.0114 | 1.9044 ±2.6849 |
| 3. | Azithromycin | J01FA10 | 2.1512 | 1.7242 | 1.3480 | 1.7411 ± 0.3281 |
| 4. | Ciprofloxacin | J01MA02 | 3.0893 | 0.0893 | 0.4314 | 1.2033 ± 1.3409 |
| 5. | Cefalexin | J01DB01 | 2.8507 | 0.0622 | 0.1406 | 1.0178 ± 1.2964 |
| 6. | Amoxicillin | J01CA04 | 0.4166 | 0.3619 | 0.6599 | 0.4794 ± 0.1295 |
| 7. | Cefixime | J01DD08 | 1.0309 | 0.0110 | 0.0080 | 0.3499 ± 0.4815 |
| 8. | Amoxicillin & beta lactamase inhibitor | J01CR02 | 0.4489 | 0.0918 | 0.3314 | 0.2907 ± 0.1486 |
| 9. | Erythromycin | J01FA01 | 0.6319 | 0.0000 | 0.0026 | 0.2115 ± 0.2973 |
| 10. | Clarithromycin | J01FA09 | 0.4666 | 0.0002 | 0.0228 | 0.1632 ± 0.2147 |
| 11. | Ampicillin | J01CA01 | 0.3030 | 0.0467 | 0.0001 | 0.1166 ± 0.1332 |
| 12. | Procaine Benzyl Penicillin | J01CE09 | 0.3490 | 0.0004 | 0.0000 | 0.1165 ± 0.1644 |
| 13. | Cefuroxime | J01DC02 | 0.1380 | 0.0000 | 0.0091 | 0.0490 ± 0.0630 |
| 14. | Sulfamethoxazole + Trimethoprim | J01EE01 | 0.0863 | 0.0044 | 0.0305 | 0.0404 ± 0.0342 |
| 15. | Metronidazole | J01XD01 | 0.0274 | 0.0169 | 0.0476 | 0.0306 ± 0.0128 |
| 16. | Cefadroxil | J01DB05 | 0.0896 | 0.0000 | 0.0000 | 0.0299 ± 0.0422 |
| 17. | Ceftriaxone | J01DD04 | 0.0368 | 0.0122 | 0.0157 | 0.0216 ± 0.0109 |
| 18. | Gentamycin | J01GB03 | 0.0619 | 0.0013 | 0.0005 | 0.0212 ± 0.0288 |
| 19. | Cefpodoxime | J01DD13 | 0.0206 | 0.0007 | 0.0014 | 0.0076 ± 0.0092 |
| 20. | Cefprozil | J01DC10 | 0.0061 | 0.0122 | 0.0036 | 0.0073 ± 0.0036 |
| 21. | Levofloxacin | J01MA12 | 0.0000 | 0.0166 | 0.0042 | 0.0069 ± 0.0070 |
| 22. | Cloxacillin | J01CF03 | 0.0064 | 0.0087 | 0.0008 | 0.0053 ± 0.0033 |
| 23. | Chloramphenicol | J01BA01 | 0.0059 | 0.0009 | 0.0019 | 0.0029 ± 0.0022 |
| 24. | Moxifloxacin | J01MA14 | 0.0000 | 0.0074 | 0.0000 | 0.0025 ± 0.0035 |
| 25. | Cefepime | J01DE01 | 0.0066 | 0.0000 | 0.0000 | 0.0022 ± 0.0031 |
| J04: Antimycobacterial | | | | | | |
| 1. | Isoniazid | J04AC01 | 0.2800 | 0.0018 | 0.0003 | 0.0940 ± 0.1315 |
| 2. | Clofazimine | J04BA01 | 0.0076 | 0.0052 | 0.0032 | 0.0053 ± 0.0018 |
| 3. | Cycloserine | J04AB01 | 0.0000 | 0.0137 | 0.0000 | 0.0046 ± 0.0065 |
| 4. | Ethambutol | J04AK02 | 0.0000 | 0.0008 | 0.0090 | 0.0033 ± 0.0041 |
| 5. | Pyrazinamide | J04AK01 | 0.0000 | 0.0061 | 0.0000 | 0.0020 ± 0.0029 |
| 6. | Rifampicin + Isoniazid | J04AM02 | 0.0000 | 0.0057 | 0.0000 | 0.0019 ± 0.0027 |
| J05 Antivirals | | | | | | |
| 1. | Lamivudine + Tenofovir + Efavirenz | J05AR11 | 1.2048 | 0.3896 | 0.7853 | 0.7932 ± 0.3329 |
| 2. | Efavirenz | J05AG03 | 0.2529 | 0.3099 | 0.0000 | 0.1876 ± 0.1347 |
| 3. | Zidovudine + Lamivudine + Abacavir | J05AR05 | 0.5319 | 0.0193 | 0.0000 | 0.1837 ± 0.2463 |
| 4. | Nevirapine | J05AG01 | 0.1995 | 0.1943 | 0.0063 | 0.1334 ± 0.0899 |
| 5. | Zidovudine + Lamivudine | J05AR01 | 0.2339 | 0.1288 | 0.0003 | 0.1210 ± 0.0955 |
| 6. | Lamivudine + Tenofovir | J05AR12 | 0.2141 | 0.0000 | 0.0019 | 0.0720 ± 0.1005 |
| 7. | Atazanavir + Ritonavir | J05AR23 | 0.0000 | 0.0001 | 0.2085 | 0.0695 ± 0.0982 |
| 8. | Tenofovir + Emtricitabine | J05AR03 | 0.0000 | 0.0000 | 0.0522 | 0.0174 ± 0.0246 |
| 9. | Lamivudine | J05AF05 | 0.0289 | 0.0233 | 0.0000 | 0.0174 ± 0.0125 |
| 10. | Abacavir | J05AF06 | 0.0228 | 0.0000 | 0.0048 | 0.0092 ± 0.0098 |

- Imported antimicrobials with incomplete data were not included in the study and may underestimate the actual consumption in Ethiopia.

- Imported and locally manufactured data at national level was used as proxy indicator for use and may not reflect the actual use of antimicrobials.

- Only antimicrobials are imported through recognized route were included in the study. However, there might be some antimicrobials that enter the country through other illegal routes and might still affect the real consumption (chiefly the expensive reserve group antibiotics).

- Only antimicrobials intended for human consumption were included in the study and will not reflect the overall antimicrobial consumption in human as well as animal health sector.

- Few antimicrobials without ATC codes and DDD values as per the WHO methodology were not included in the survey.

Moreover, in Ethiopia, there was wide variation in quantity, type of antimicrobials imported each year, and makes drawing conclusions challenging to say the consumption trend was either increasing or decreasing.

In spite of the limitations, this study estimated and identified the trends of consumption of antimicrobials over three years and the most frequently consumed antimicrobials in Ethiopia. Moreover, the study identified the need to improve documentation systems for imported antimicrobials, in terms of completeness and accuracy of data currently collected.

## Conclusion

National antimicrobial consumption for Ethiopia was estimated at 15.36 DID between 2017 and 2019 and was between the WHO global comprehensive antimicrobial consumption range of 4.4–64.4 DID. Majority (98.6%) of the antimicrobial consumption was orally administered antimicrobials. Nearly 3/4th of antimicrobials consumed were in the Access AWaRe category of Ethiopia AWaRe Classification. Moreover, doxycycline, norfloxacin, azithromycin and ciprofloxacin accounted for nearly 70% of the overall antimicrobial consumption. Hence, policy makers, regulators, professional associations, prescribers, dispensers and the community should be curious about the rational use of these group antimicrobials. Finally, we strongly recommend further derailed studies at national, regional and facility level be conducted by including animal health sector as well.

## Acknowledgments

The authors are thankful to professionals working at all the recognized entry ports and the local manufacturers for their cooperation during the study.

## Author Contributions

**Conceptualization:** Million Tirfe, Asnakech Alemu, Wondie Alemu, Mengistab Woldearegay, Getachew Asfaw, Duru Kadi, Atalay Mulu Fentie.

**Data curation:** Million Tirfe, Asnakech Alemu, Wondie Alemu, Mengistab Woldearegay, Getachew Asfaw, Duru Kadi.

**Formal analysis:** Million Tirfe, Asnakech Alemu, Wondie Alemu, Getachew Asfaw, Atalay Mulu Fentie.

**Funding acquisition:** Million Tirfe, Asnakech Alemu, Wondie Alemu, Heran Gerba.

**Investigation:** Asnakech Alemu, Wondie Alemu, Mengistab Woldearegay, Getachew Asfaw, Duru Kadi.

**Methodology:** Million Tirfe, Asnakech Alemu, Wondie Alemu, Mengistab Woldearegay, Getachew Asfaw, Heran Gerba, Duru Kadi, Atalay Mulu Fentie.

**Project administration:** Million Tirfe, Wondie Alemu, Mengistab Woldearegay, Heran Gerba.

**Resources:** Million Tirfe, Wondie Alemu, Heran Gerba.

**Software:** Asnakech Alemu, Atalay Mulu Fentie.

**Supervision:** Getachew Asfaw, Heran Gerba, Atalay Mulu Fentie.

**Validation:** Asnakech Alemu, Getachew Asfaw, Duru Kadi, Atalay Mulu Fentie.

**Visualization:** Million Tirfe, Asnakech Alemu, Wondie Alemu, Mengistab Woldearegay, Getachew Asfaw, Heran Gerba, Duru Kadi, Atalay Mulu Fentie.

**Writing – original draft:** Million Tirfe, Asnakech Alemu, Wondie Alemu, Mengistab Woldearegay, Getachew Asfaw, Duru Kadi.

**Writing – review & editing:** Million Tirfe, Asnakech Alemu, Wondie Alemu, Mengistab Woldearegay, Getachew Asfaw, Heran Gerba, Duru Kadi, Atalay Mulu Fentie.

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
