## [Decision Letter · Decision Letter 0]

20 Feb 2023

PONE-D-22-35443A three years antimicrobials consumption in Ethiopia from 2017 to 2019: A cross- sectional studyPLOS ONE

Dear Dr. Fentie,

Thank you for submitting your manuscript to PLOS ONE. After careful consideration, we feel that it has merit but does not fully meet PLOS ONE’s publication criteria as it currently stands. Therefore, we invite you to submit a revised version of the manuscript that addresses the points raised during the review process.

 ==============================There are minor changes to be made on the manuscript.==============================

We look forward to receiving your revised manuscript.

Kind regards,

Arianit Jakupi, PhD

Academic Editor

PLOS ONE

Journal Requirements:

2. In your Methods section, please confirm that all data sources you used were publicly available and anonymized. If this is not the case, please provide information on what permissions you were granted to access these data.

"No. The funders had no role in study design, data collection and analysis, decision to publish, or preparation of the manuscript"

Reviewers' comments:

Reviewer's Responses to Questions

**Comments to the Author**

1. Is the manuscript technically sound, and do the data support the conclusions?

Reviewer #1: Yes

Reviewer #2: Yes

2. Has the statistical analysis been performed appropriately and rigorously? 

Reviewer #1: Yes

Reviewer #2: Yes

3. Have the authors made all data underlying the findings in their manuscript fully available?

Reviewer #1: Yes

Reviewer #2: Yes

4. Is the manuscript presented in an intelligible fashion and written in standard English?

Reviewer #1: Yes

Reviewer #2: Yes

5. Review Comments to the Author

Reviewer #1: In this manuscript, the authors have evaluated a three years antimicrobials consumption (imported and locally manufactured antimicrobials), as a retrospective cross-sectional investigation, in Ethiopia from 2017 to 2019, respectively. The rationale of manuscript is very good. The paper is well written and clearly organized. The authors have done a good job of presenting their work and clearly spent some effort on the editing, which has resulted in a paper that is easy to read.

I have a few comments, but no major changes are suggested.

1. There should be a space between the word after or before it and all the references should be the same, a space before it or not.

2. At the table 4. You have present most commonly consumed antimicrobials by chemical substance group in Ethiopia between 2017-2019. Is there any explanation why is such a big difference at DID between the year 2017 and two other years (2018 and 2019)?

3. A number of similar studies have been published in past by a number of research groups. Moreover, current findings are also not conceptually and technically different from the earlier published studies. Due to the large sample, my recommendation is to the authors that they should emphasize and elaborate the novel findings and speculate what are the contribution of this study to previously published studies.

Reviewer #2: Overall the article is written in a very comprehensive way. Chosen methodology is correct. Using DID gives comparable results and data for further drug utilisation studies.

However it should be mentioned that it should be good to have also results that for the most consumed antibiotics in order to create a link with the resistance as well. One other suggestion is also to describe the fluctuations of different classes of antibiotics through the years and if there is supporting evidence in prescriptions of these antibiotics.

6. PLOS authors have the option to publish the peer review history of their article (what does this mean?). If published, this will include your full peer review and any attached files.

Reviewer #1: **Yes: **Rozafa Koliqi

Reviewer #2: No

---

## [Author Response · Author response to Decision Letter 0]

6 Mar 2023

Date: March 7, 2023

Response Letter to the Editor and Reviewers 

We appreciate the editor and both reviewers’ time, comments and suggestions provided for our article entitled “A three years antimicrobials consumption in Ethiopia from 2017 to 2019: A cross- sectional study: Manuscript ID= PONE-D-22-35443”. Reviewer comments motivated thorough revision of the manuscript, which is being resubmitted along with a point-by-point response to address all reviewer concerns (Table below). All modifications are shown in the revised manuscript attached with file name of “Manuscript” for manuscript without track change and “Revised Manuscript with track changes” for the file highlighted with changes. Thank you for the valuable comments and we hope the editor and reviewers will be satisfied with our responses. 

Sincerely,

Atalay Mulu Fentie, on behalf of all authors

S.No Query by Response 

1. Editor 

1. 1.1. Please ensure that your manuscript meets PLOS ONE's style requirements, including those for file naming. All authors have thoroughly reviewed the manuscript and made changes as required to meet all of the PLOS ONE style requirements. In addition, all authors addressed identified grammatical and typographic errors (see the revised manuscript with track changes). 

2. 1.2. In your Methods section, please confirm that all data sources you used were publicly available and anonymized. If this is not the case, please provide information on what permissions you were granted to access these data. Thank you so much. Now, we have included “All data sources used in this study were publicly available and anonymized” under source of data Methods sub-section of the revised manuscript.

3. 1.3. Thank you for stating the following financial disclosure: 

"No. The funders had no role in study design, data collection and analysis, decision to publish, or preparation of the manuscript"

Please include your amended statements within your cover letter; we will change the online submission form on your behalf. Now we have the final disclosure section as per your recommendations and on the revised manuscript it appears as follows:

“This national survey was financially supported by Ethiopian Food and Drug Authority through the annual government budget allocated specifically for the consumption survey. However, the funder had no role in study design, data collection and analysis, decision to publish, or preparation of the manuscript. In connection with this, the authors received no specific funding for this work.”

4. 1.4. In your Data Availability statement, you have not specified where the minimal data set underlying the results described in your manuscript can be found. We have amended the data availability statement as follows and included it on the revised manuscript. “Minimum data relevant to this study are anonymized and included in this article”.

5. 1.5. Please review your reference list to ensure that it is complete and correct. We have revised the reference section as per PLOS ONE reference writing guideline, and we hereby confirm it is complete and correct. 

2. 2. Reviewer 1- Dr. Rozafa Koliqi 

2.1. There should be a space between the word after or before it and all the references should be the same, a space before it or not. Thank you. As per the kind comment, we have thoroughly reviewed the manuscript and amended it accordingly. 

2.2. At the table 4. You have present most commonly consumed antimicrobials by chemical substance group in Ethiopia between 2017-2019. Is there any explanation why is such a big difference at DID between the year 2017 and two other years (2018 and 2019)? It could be due to the acute watery diarrheal outbreak in Ethiopia in 2017 that leads the consumption of antimicrobials high. If you look at the consumption of Doxycycline, it was significantly higher given doxycycline was recommended on the emergency preparedness and management of acute diarrheal guideline as first line treatment. We have included the same on the discussion section of the manuscript on paragraph 4. 

2.3. A number of similar studies have been published in past by a number of research groups. Moreover, current findings are also not conceptually and technically different from the earlier published studies. Due to the large sample, my recommendation is to the authors that they should emphasize and elaborate the novel findings and speculate what are the contribution of this study to previously published studies. Thank you for this comment and yes, recently we have developed the AMR prevention and containment national action plan of Ethiopia. In line to this, one of the strategic objective is research and surveillance to support all other intervention strategies and policies in which this study was conducted in accordance with it. Moreover, we have revised the Essential medicine list, standard treatment guideline, health insurance medicine list and community pharmacy list; and categorized antibiotics as per Access, Watch and Reserve (AWaRe). We tried discuss and present the antimicrobial consumption as per AWaRe category that will serve a baseline for further study as well as assess the five years national action plan impact on antimicrobial consumption and AMR. 

3. Reviewer 2 

3.1. Overall the article is written in a very comprehensive way. Chosen methodology is correct. Using DID gives comparable results and data for further drug utilisation studies. However it should be mentioned that it should be good to have also results that for the most consumed antibiotics in order to create a link with the resistance as well. One other suggestion is also to describe the fluctuations of different classes of antibiotics through the years and if there is supporting evidence in prescriptions of these antibiotics. Thank you for your suggestion. We tried to highlight the findings of studies conducted in Ethiopia regarding to antimicrobial resistance pattern/issue on the discussion section of the manuscript. Moreover, the reasons for fluctuation of the consumption was also discussed. The only supporting evidence we found is the issue of acute watery diarrhea in 2017 that significantly increased the consumption compared to 2018 and 2019.

---

## [Editor Report · Decision Letter 1]

21 Mar 2023

A three years antimicrobials consumption in Ethiopia from 2017 to 2019: A cross- sectional study

PONE-D-22-35443R1

Dear Dr. Fentie,

We’re pleased to inform you that your manuscript has been judged scientifically suitable for publication and will be formally accepted for publication once it meets all outstanding technical requirements.

Kind regards,

Arianit Jakupi, PhD

Academic Editor

PLOS ONE
---

## [Editor Report · Acceptance letter]

28 Mar 2023

PONE-D-22-35443R1 

A three years antimicrobials consumption in Ethiopia from 2017 to 2019: A cross- sectional study 

Dear Dr. Fentie:

I'm pleased to inform you that your manuscript has been deemed suitable for publication in PLOS ONE. Congratulations! Your manuscript is now with our production department. 

Kind regards, 

on behalf of

Dr Arianit Jakupi 

Academic Editor

PLOS ONE